



# Atmospheric mercury in the southern hemisphere tropics: seasonal and diurnal variations and influence of inter-hemispheric transport

Dean Howard[1], Peter F. Nelson[1], Grant C. Edwards[1], Anthony L. Morrison[1], Jenny A. Fisher[2,3], Jason Ward[4], James Harnwell[4], Marcel van der Schoot[4], Brad Atkinson[5], Scott D. Chambers[6], Alan D. Griffiths[6], Sylvester Werczynski[6], and Alastair G. Williams[6]

[1]Department of Environmental Sciences, Macquarie University, Sydney, New South Wales, 2109, Australia
[2]Centre for Atmospheric Chemistry, School of Chemistry, University of Wollongong, Wollongong, New South Wales, 2552, Australia
[3]School of Earth & Environmental Sciences, University of Wollongong, Wollongong, New South Wales, 2552, Australia
[4]Oceans and Atmosphere Flagship, Commonwealth Science and Industrial Research Organisation, Aspendale, Victoria, 3195, Australia
[5]Darwin Research Station, Bureau of Meteorology, Darwin, Northern Territory, 0810, Australia
[6]Institute for Environmental Research, Australian Nuclear Science and Technology Organisation, Sydney, New South Wales, 2232, Australia

*Correspondence to:* Dean Howard (dean.howard@mq.edu.au)

**Abstract.** Mercury is a toxic element of serious concern for human and environmental health. Understanding its natural cycling in the environment is an important goal towards assessing its impacts and the effectiveness of mitigation strategies. Due to the unique chemical and physical properties of mercury, the atmosphere is the dominant transport pathway for this heavy metal, with the consequence that regions far removed from sources can be impacted. However, there exists a dearth of long-term mon-

5 itoring of atmospheric mercury, particularly in the tropics and southern hemisphere. This paper presents the first two years of gaseous elemental mercury (GEM) measurements taken at the Australian Tropical Atmospheric Research Station (ATARS) in northern Australia, as part of the Global Mercury Observation System (GMOS). Annual mean GEM concentrations determined at ATARS ($0.95 \pm 0.12$ ng m$^{-3}$) are consistent with recent observations at other sites in the southern hemisphere. Comparison with GEM data from other Australian monitoring sites suggests a concentration gradient that decreases with increasing lati-

10 tude. Seasonal analysis shows that GEM concentrations at ATARS are significantly lower in the distinct wet monsoon season than in the dry season. This result provides insight into alterations of natural mercury cycling processes as a result of changes in atmospheric humidity, oceanic/terrestrial fetch and convective mixing, and invites future investigation using wet mercury deposition measurements. Due to its location relative to the atmospheric equator, ATARS intermittently samples air originating from the northern hemisphere, allowing an opportunity to gain greater understanding of inter-hemispheric transport of mercury

and other atmospheric species. Diurnal cycles of GEM at ATARS show distinct nocturnal depletion events that are attributed to dry deposition under stable boundary layer conditions. These cycles provide strong further evidence for the "multi-hop" model of global GEM cycling, whereby long-range transport is characterised by multiple surface depositions and re-emissions, rather than continuous transport over long distances.



# 1 Introduction

Mercury (Hg) is a toxic element that has natural and anthropogenic sources, sinks and cycles within the environment. Human activities such as gold mining and biomass/fossil fuel combustion have perturbed the natural cycling of mercury through the addition of mercury emissions, which are re-deposited from the atmosphere to land, vegetation and water bodies. It is estimated that currently anthropogenic sources increase the global atmospheric mercury pool by 1960 tonnes annually, a value that represents 30 % of estimated mercury emissions, with the remainder emitted from natural geological sources (10 %) or re-emitted from stores of previously-deposited mercury (60 %). These mercury emission estimates are subject to large uncertainties (AMAP/UNEP, 2013; UNEP, 2013). That anthropogenic mercury sources now exceed those from natural sources on a global scale is of concern for both human and environmental health. Evidence suggests these additional sources are leading to increased concentrations of mercury in the oceans and in marine animals, with the consequence that bioaccumulation of toxic methylmercury within aquatic food chains has also increased (Mason et al., 2012; UNEP, 2013). There exists a significant pathway for methylmercury transfer to humans, as it is estimated that more than 100 million tonnes of fish are eaten world-wide each year and fish provide two and a half billion people with at least 20 % of their protein intake. Mercury in this latter form can seriously threaten human health through impacts on the development of foetuses and young children. In response to this threat, the United Nations Environment Programme (UNEP) has developed the Minamata Convention on Mercury, which is expected to be ratified in 2017.

The global cycling of mercury is unique amongst metals, resulting from its existance primarily as a gas at ambient environmental conditions (Edwards et al., 2001). Within Earth's atmosphere, 90 to 99 % of mercury is found as gaseous elemental mercury (GEM), with the remaining portion composed of operationally-defined gaseous oxidised mercury (GOM) and particulate-bound mercury (PBM) — collectively known as reactive mercury (RM) (Gustin et al., 2013). The low atmospheric reactivity and extremely low solubility of the elemental form (GEM) results in low wet/dry deposition rates and scavenging of GEM from the atmosphere, with consequential long atmospheric lifetimes estimated at 6–12 months contributing to long-range transport of this species (Holmes et al., 2006; Selin et al., 2007; Holmes et al., 2010). These attributes result in atmospheric transport being the dominant distribution mechanism through the environment, with long-range transport possible across hemispheric scales. Differences in background atmospheric mercury concentrations between the hemispheres are hence dependent on emission rates, atmospheric mercury lifetimes, deposition rates, and inter-hemispheric transport processes.

With 68 % of the Earth's landmass and 88 % of the human population in the northern hemisphere, both natural and anthropogenic emissions of mercury are disproportionately distributed between the hemispheres. Towards the equator, the existence of the intertropical convergence zone (ITCZ) and the associated upward/poleward movement of the Hadley circulation leads to reduced tropospheric mixing across the atmospheric, or chemical equator (Bowman and Cohen, 1997; Hamilton et al., 2008; Holmes and Prather, in press) and hence a broad, hemispheric gradient of GEM concentrations (Sprovieri et al., 2016). Stationary observations of GEM within the tropics are rare but those that are available report significant changes in concentration as source regions shift across hemispheres with the drift of the atmospheric equator (Müller et al., 2012; Wang et al., 2014). The tropics also represent an important region for mercury cycling as they are home to around 40 % of the world's population, in-



cluding over 50 % of people under the age of 15, a group at greater risk of adverse effects due to mercury exposure during early development (Bose-O'Reilly et al., 2010). Furthermore, this region hosts several large coastal communities within emerging and developing economies, in which environmental controls and advisories are not always well developed (Costa et al., 2012).

Characterisation of background GEM in the tropics and southern hemisphere (SH) has been hindered by a lack of observa-
tions and is based largely on intermittent ship voyages (Soerensen et al., 2012), along with a few long-term stationary records in South America, Africa, Antarctica, and islands in the Indian and eastern Pacific oceans (Sheu et al., 2010; Müller et al., 2012; Wang et al., 2014; Angot et al., 2014; Slemr et al., 2015; Angot et al., 2016). A recent comparison of interannual records from four mercury monitoring stations spanning a latitude range of 34° S to 72° S, of which the longest-running spans 7 years, suggests that background GEM concentrations in the southern hemisphere are between 0.85 and 1.05 ng m$^{-3}$ (Slemr et al., 2015).
Previous measurements of atmospheric mercury concentrations have also been reviewed by Sprovieri et al. (2010, 2016). The Australian continent, with its large non-Antarctic southern hemisphere landmass (22 %), a latitudinal distribution (11–44° S) spanning diverse climatic zones, and a mercury emission profile characterised by anthropogenic sources that are significantly smaller than natural and re-emitted sources (Nelson et al., 2012), presents unique opportunities for extending environmental mercury monitoring in a region that has largely been under-represented.

Initiated under the Global Mercury Observation System (GMOS) and considered for inclusion with the Asia Pacific Mercury Monitoring Network (APMMN), measurements of GEM are being undertaken at the Australian Tropical Atmospheric Research Station (ATARS), north-east of Darwin in Australia's Northern Territory. Of the six GMOS sites classed as tropical, ATARS is the southernmost and one of only two (along with Kodaicanal; 10.2314° N, 77.4652° E) situated in the eastern hemisphere. This site is therefore important in bridging the spatial gap in GEM measurements in equatorial regions around the globe.
Originally an experimental radar site, ATARS was expanded in 2010 to incorporate greenhouse gas measurements as part of the Australian Greenhouse Gas Observation Network (Ziehn et al., 2016) and is operated jointly by the Australian Bureau of Meteorology (BoM) and the Commonwealth Science and Industrial Research Organisation (CSIRO). The Australian Nuclear Science and Technology Organisation (ANSTO) began continuous atmospheric radon measurements at the site in 2012 to aid in the determination of terrestrial influence on observed air masses (Chambers et al., 2016b). In June 2014, an additional
expansion took place and now continuous aerosol, reactive gas (O$_3$, NO$_x$) and GEM measurements complement the suite of atmospheric measurements at the site (Mallet et al., 2016). This GEM dataset represents the first multi-year time series of atmospheric mercury monitoring in tropical Australia.

We present here the first two years of tropical GEM measurements from ATARS, examine their seasonal and diurnal variations, and evaluate the contribution of air masses transported from the northern hemisphere to the observed concentrations.
These results add substantial new information to our understanding of mercury in the southern hemisphere and tropical atmosphere.



## 2 Methods

### 2.1 Site description

ATARS is situated on the Gunn Point peninsula (12.2491° S, 131.0447° E; Fig. 1), approximately 20 km north-east from the suburban edge of Darwin (2013 population 136,200; ABS, 2015) in Australia's Northern Territory. Between 2 and 9 km to the

north and west of ATARS lies the edge of the peninsula that gives way to the Tiwi Islands and Timor Sea, whilst the land to the east and south is largely uninhabited and includes national parks and conservation areas.

The climate in the region is best described as tropical (Köppen Aw; Peel et al., 2007) with mean monthly maximum temperatures between 30 °C and 33 °C (1941 to 2016 means; BoM, 2016) and a distinct monsoon (wet) season that coincides generally with the austral summer (December–February). The build-up to these monsoon seasons is characterised by steadily

increasing minimum temperatures (19 °C in July to 25 °C in December) and associated increases in humidity (daily ranges of 37–60 % in July to 72–83 % in February). Mean annual rainfall is 1728 mm, with an average of 1604 mm (>90 %) of this falling in the period November–April. As the site is located on a peninsula, a sea/land breeze cycle is often experienced in the dry season, resulting in mostly south-easterly winds throughout the morning, tending northerly as the sea breeze circulation sets in from the nearby coast. In the wet season, shifting synoptic patterns result in an increased frequency of westerly winds.

The vegetation classification is savannah with coarse grasses and scattered tree growth.

Anthropogenic emissions of mercury and its compounds to the atmosphere in and around Darwin are generally quite low. Australian National Pollutant Inventory (NPI) data for 2014–15 state that 6 sites situated between 20 km and 40 km from ATARS in the direction of Darwin (wind directions 190° to 240°) emitted a total of 0.12 kg Hg to the atmosphere (NPI, 2016). Other distributed anthropogenic mercury emissions in Darwin are estimated at less than 0.2 kg a$^{-1}$, based on 25 km x 25 km

gridded population data (Nelson et al., 2012).

### 2.2 Measurements

Continuous (5-minute sample) GEM measurements were obtained using a Tekran 2537X Automated Ambient Air Analyser (2537X). This instrument is housed in an air-conditioned structure with internal temperature set at 25 °C. Air is sampled from a 10 m high tower through 7.95 mm I.D. perfluoroalkoxy tubing using a Thomas 2688 vacuum pump drawing approximately

50 l min$^{-1}$ (residence time 0.6 s). The 2537X subsamples from this flow at 1 l min$^{-1}$ through 6 m of heated polytetrafluoroethylene (PTFE) line maintained at 50 °C, and two 0.2 $\mu$m PTFE filters positioned before and after the heated line. The 2537X operates on the principle of cold vapour atomic fluorescence spectroscopy (CVAFS) following gold amalgamation preconcentration (see for example Ebinghaus et al., 1999; Munthe et al., 2001). This technique quantifies total gaseous mercury (TGM = GEM + GOM), however experience from other researchers suggests that the fraction of GOM in the atmosphere is

generally small and removed upstream of the 2537X. As such we present the results here as GEM and not TGM, in line with reporting standards employed by other GMOS secondary sites (Sprovieri et al., 2016).

Quality assurance and quality control procedures were applied as per protocols derived for GMOS sites (Sprovieri et al., 2016). Calibration of the 2537X took place every 23 hours using an internal mercury permeation source maintained at 50 °C.



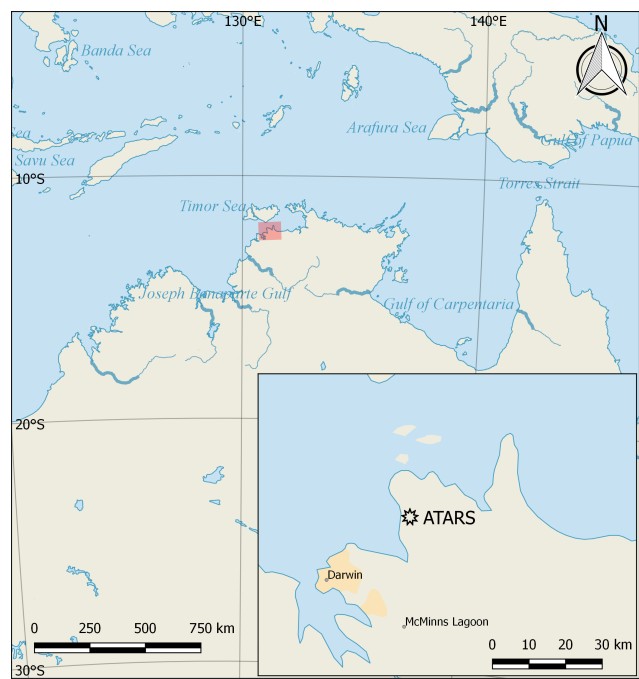

**Figure 1.** Map of region surrounding ATARS. Composed in QGIS using Natural Earth dataset.

Primary calibration of this source took place twice each year using manual injections of mercury vapour. No change in the internal source permeation rate was detected over this period. Furthermore, standard additions of mercury are automatically introduced to the 2537X from the internal permeation source every 35 samples (∼3 hours) in order to verify GEM recovery performance.

Continuous hourly measurements of radon were sampled at 12 m using an ANSTO-designed and -built, 700 l dual-flow-loop two-filter radon detector (Whittlestone and Zahorowski, 1998; Chambers et al., 2011). This detector samples at $40\,\mathrm{l\,min^{-1}}$ through 25 mm high-density polyethylene agricultural pipe and has a lower limit of detection of 40–50 mBq m$^{-3}$. Calibrations are performed monthly by injecting radon from a 101.15 ± 4 % kBq $^{226}$Ra source (delivering 12.745 Bq $^{222}$Rn min$^{-1}$), traceable to NIST standards. Instrumental background is checked every 3 months. Radon measurements were corrected for

the response time of the instrument (Griffiths et al., 2016), although the main trends were not affected by this time correction. Time-corrected radon data were then split into "fetch" and "diurnal" components by interpolating between minimum afternoon (12:00 to 17:00) values when atmospheric mixing is greatest and subtracting these interpolated values (fetch component) from the original signal, leaving the diurnal component (see Chambers et al., 2016a, for details).

    Meteorological measurements are collected at ATARS using a standard automated weather station (AWS) operated by the

Australian Bureau of Meteorology. Precipitation data were collected using a 203 mm tipping bucket rain gauge and daily totals were summed to give cumulative season totals centred around a hydrologic year beginning 1st June. The temporal extents of what we define here as "wet seasons" were then determined using the method of Smith et al. (2008), whereby 15 % and 85 % of





the total cumulative rainfall marked their onset and conclusion, respectively. The wet season of 2014–15 was further extended to include two 100+ mm rain events that took place in November and March.

## 2.3 Modelling

As the atmospheric equator changes its position relative to the geographic equator, we employed a system of passive tracers within the GEOS-Chem chemical transport model to help assess the impact of air originating from the northern hemisphere (NH) on the site, based on the work of Holmes and Prather (in press). We use GEOS-Chem v10-01 driven by assimilated meteorology from the NASA Goddard Earth Observing System Forward Processing (GEOS-FP) data product, run at 2° x 2.5° horizontal resolution and 47 vertical levels from the surface to 0.01 hPa. Tracers with 90-day lifetimes were uniformly released from the surface in all model boxes poleward of 45° latitude within each hemisphere. The atmospheric equator is then defined as the point where mixing ratios of tracers from the two hemispheres are equal. Tracer concentrations in surface air over ATARS were saved as daily mean values in the model grid box containing the site (2° latitude by 2.5° longitude and an approximate atmospheric depth of 130 m). Increasing the number of grid squares over which tracer values were averaged did not significantly affect the results.

The NOAA Hybrid Single Particle Lagrangian Integrated Trajectory (HYSPLIT) Model (Draxler, 1999; Draxler and Hess, 1998; Stein et al., 2015) was also employed to assess influences of air mass source regions. Global Data Assimilation System (GDAS) 0.5° meteorological reanalysis data were used to drive the model, and trajectories were initialised at 0.5 times the mixed layer height as determined by HYSPLIT. To reduce the influence of local daily variation in GEM concentrations on this analysis, back trajectories were calculated for each hour of the day rather than as a daily or part-daily mean. For each trajectory, air parcel coordinates were calculated every two hours and weighted per the corresponding GEM concentration. These weighted values were then averaged over 0.5° x 0.5° grid cells.

## 3 Results and discussion

### 3.1 Overall means and seasonal trends

Measurements of GEM at ATARS began on 5th June, 2014 and were still ongoing at the time of writing. Instrument maintenance/downtime plus application of QC protocols, including calibration and standard additions, resulted in 68.1 % temporal measurement coverage during the first two years of operation (Fig. 2, Table 1). Concentrations are normally distributed across this period with an overall mean of 0.95 ± 0.12 ng m$^{-3}$ (1 standard deviation), which is within the range of long-term background GEM concentrations for the southern hemisphere as reported by Slemr et al. (2015). Mean GEM concentrations reported by Slemr et al. (2015) over 2012–13 at Cape Grim, Tasmania (40.6832° S, 144.6899° E) and by Morrison et al. (2015) over 2014–15 at Singleton, NSW (32.4777° S, 151.1018° E) were both 0.86 ng m$^{-3}$ (9 % lower), suggesting a slight latitudinal gradient in GEM across the Australian continent. A latitudinal gradient within the southern hemisphere was also more generally seen in median annual GEM concentrations for GMOS sites in 2013–14, based on data from 5 sites (Sprovieri





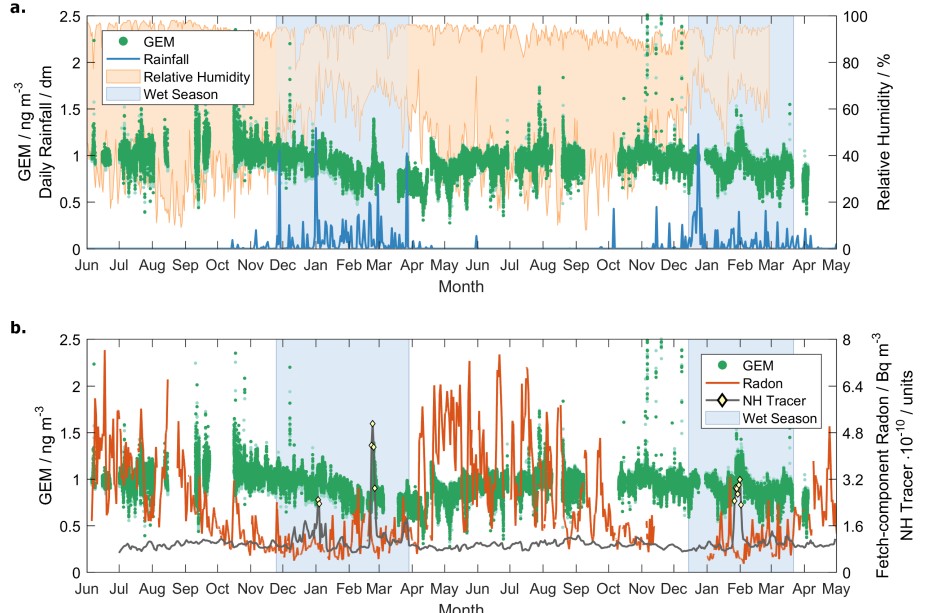

**Figure 2. a.** 5-minute GEM data, daily rainfall and daily min/max relative humidity values plus wet season ranges as defined by Smith et al. (2008). **b.** 5-minute GEM, hourly fetch-component radon and daily NH tracers. Days defined as NH-influenced are marked with diamonds.

et al., 2016). GEM measurements at ATARS were coincident with those reported by Sprovieri et al. (2016) for only the latter 6 months of 2014, a period spanning the late dry season and early wet season. Concentrations during this period were $1.02 \pm 0.10$ ng m$^{-3}$ — higher than the overall mean at ATARS, though still lower than mean values reported for other tropical GMOS sites.

A seasonal trend is apparent in the GEM time series (Fig. 2), which shows higher concentrations during the dry season compared to the wet. Wind sector analysis also shows distinctly different wind patterns between wet and dry seasons (Fig. 3). During the wet season, $\sim$60 % of winds come to the site from a westerly direction, consistent with shifting of the ITCZ and associated low pressure systems towards northern Australia. In the dry season, south-easterly to north-easterly winds are more common ($\sim$65 % between 30° and 150°), although there is also a notable westerly element. Concentration distributions vary

between seasons, with a larger fraction of values above 1 ng m$^{-3}$ seen in the dry period. Within each season however, these distributions do not change significantly with wind direction. Furthermore, the small percentage of winds arriving from the south-west show no change in GEM distribution, implying that the low mercury emissions from Darwin are not significantly impacting measurements and that overall trends are indicative of influences from the global atmospheric mercury pool rather than local sources.

Figure 2 shows that the highest GEM values are concentrated into short peaks, clustered more heavily around the mid- to late-dry season. In the absence of local anthropogenic sources, this is considered consistent with biomass burning events and the associated release of mercury from volatilisation and thermal desorption from vegetation and soils (Melendez-Perez et al.,



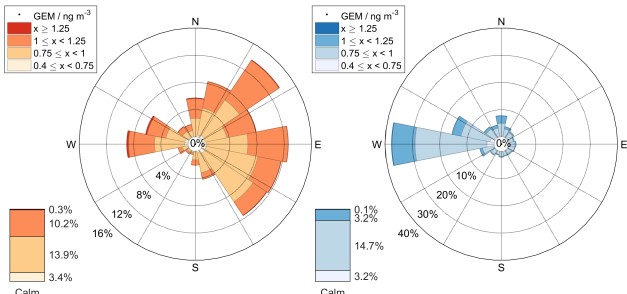

**Figure 3.** Directional GEM concentration distributions for (left) dry season and (right) all wet season half-hourly GEM data.

2014). These biomass burning events occur extensively in Northern Australia throughout the dry season as the result of natural and accidental lighting, as well as part of local land management practices (Russel-Smith et al., 2007). Nelson et al. (2009) concluded that burning in the northernmost part of Australia can contribute up to around 2 kg Hg km$^{-2}$ a$^{-1}$ to the atmosphere (2006 data, 25 km x 25 km grid resolution).

An intensive study of these biomass burning events undertaken at ATARS during the early dry season in 2014 also confirmed spikes in GEM concentration that were associated with biomass burning (Mallet et al., 2016; Desservettaz et al., in press). The distance to the fire and atmospheric dispersion, as well as vegetation type and associated mercury loading, were all identified as factors influencing the strength of these biomass burning signals. Desservettaz et al. (in press) calculated emission factors for GEM between 0.0035 and 0.032 g Hg per kg dry fuel, around 2 orders of magnitude higher than that determined by Andreae and

Merlet (2001) over savannah grasslands. The fires observed by Desservettaz et al. were shown to be from scrubland fires rather than grassland fires, excluding the possibility of direct comparison between the two results. With a full suite of greenhouse gas and aerosol measurements taking place at ATARS, further identification of smoke plumes and precise calculation of emission factors is possible in a manner that is comparable with previous studies.

      Wet season GEM concentrations in 2014–15 were characterised by a steady, gradual decrease that reversed abruptly in early

April shortly after the onset of the dry season (Fig. 2). GEM concentrations during the 2015–16 wet season saw a similar, though much less distinct decrease over a shorter and drier season. Figure 2 also shows that fetch-component radon concentrations begin to drop in both years around September–October, which HYSPLIT trajectories show is coincident with air mass origin shifting away from the Australian continent and towards the northern Arafura and Timor Seas. Throughout the wet season fetch-component radon remains low, though not at baseline levels (Zahorowski et al., 2013; Chambers et al., 2016b), implying

that there is still some terrestrial influence on incoming air masses from the Australian continent or surrounding islands to the north. Wet season wind data (Fig. 3) confirm that the predominant fetch during this period is from the west, where the Timor Sea lies less than 2 km from ATARS. Air-sea exchange of GEM is complex, although the ocean is generally considered a sink for atmospheric mercury (Mason and Sheu, 2002; Song et al., 2015). Costa et al. (2012), using GEOS-Chem, modelled a total mercury deposition flux of 10 to 12 $\mu$g m$^{-2}$ a$^{-1}$ in the Timor Sea region, whereas Nelson et al. (2012) modelled

terrestrial mercury emission fluxes over Australia that were generally between 8 and 44 $\mu$g m$^{-2}$ a$^{-1}$ from soil and vegetation.





The increase in GEM concentrations in the early 2015 dry season was coincident with a shift to largely terrestrial-influenced fetch, as evidenced by a coincident increase in fetch-component radon, however the timing offset between decreases in GEM and fetch-component radon in the early wet and late dry seasons suggests that air mass origin is not the only influence on wet season GEM decreases.

Within tropical regions, wet deposition has been shown to be a significant pathway for mercury from the atmosphere to ecosystems, even in relatively low-mercury air and despite the low solubility of mercury in its elemental form (Fostier et al., 2000; Costa et al., 2012; Hansen and Gay, 2013; Shanley et al., 2015). mercury "rainout" — or the tendency for mercury rainwater loading to decrease with increasing precipitation — has also been demonstrated in Mercury Deposition Network (MDN) data in North America (Glass and Sorensen, 1999; Prestbo and Gay, 2009) and positive correlations between GEM

(TGM) and rainwater mercury have been reported in MDN data (GEM; Cole et al., 2014) and at Cape Point, South Africa (TGM; Brunke et al., 2016). Furthermore, increases in minimum relative humidity during the wet season may have an impact on aqueous oxidative chemistry, increasing the likelihood that GEM may be converted to the more reactive GOM or PBM and subsequently deposited to the surface. Mercury wet deposition is currently not being measured at ATARS; however, given the large differences in GEM trends between the wet and dry seasons, deposition measurements could help to highlight differing

processes between these periods.

### 3.2  Daily variation

Short, significant troughs in GEM values can be seen in Fig. 2, down to a minimum value of 0.28 ng m$^{-3}$. These are more pronounced in the dry season, though still common during the wet. GEM recoveries from standard additions during these periods were investigated and remained within 10 % of expected values with no evident pattern throughout the day, implying

the drops in observed GEM were due to natural phenomena and not a change in instrument GEM recovery. Atmospheric mercury depletion events (AMDEs) and the mechanisms behind them have been well-documented in polar regions (Steffen et al., 2008), though other similar events have been observed within the mid-latitudes (Mao et al., 2008; Brunke et al., 2010; Engle et al., 2010; Moore et al., 2013; Morrison et al., 2015; Howard et al.). The mechanisms behind these mid-latitude depletion events are less clear and likely varied, with hypotheses such as chemical conversion of GEM to RM and subsequent

deposition; transport of GEM-depleted air masses; or deposition of GEM from isolated atmospheric pools, being offered. Closer inspection of the dips in GEM observed at ATARS reveals that they occur overnight and are particularly pronounced in the early hours of the morning, with a marked rebound following sunrise. Previous flux studies have shown that surface GEM fluxes over soils with mercury concentration at background levels (such as at ATARS, with mean soil mercury concentration of 18 $\mu$g kg$^{-1}$) are generally bi-directional, with little controlling influence from soil mercury concentration (Agnan et al., 2016,

and references within). Correlations with solar radiation and air temperature tend to lead to emission fluxes throughout the day and deposition or near-zero flux overnight over terrestrial surfaces (Edwards and Howard, 2013).

The pattern of overnight GEM depletion is shown in diurnal composite data in Fig. 4, along with the diurnal-component of radon and wind direction. Days have been defined from midday to midday, then sorted into groups according to quartiles of the radon diurnal-component at sunrise (marked in the top figures). As radon fluxes are, across daily timescales, constant





to first-order approximation, nocturnal build-up of radon is indicative of atmospheric stability, with highest radon values indicating the most stable atmospheres. This follows the radon-based stability categorisation method described by Chambers et al. (2016a) and Williams et al. (2016). In the dry season (left), it can clearly be seen that the magnitude of nocturnal GEM depletion increases with increasing stability, and conversely, little-to-no depletion occurs under well-mixed boundary layers.

Wind directions for the well-mixed category shift from coastal (westerly) in the early evening to terrestrial during the night. In contrast, wind directions for moderately-mixed to stable boundary layer categories are very similar to each other, shifting from a north-easterly to south-easterly direction shortly after sunset. Terrestrial fetches encompass this range of directions and the abrupt shift in wind direction has little impact on the rates of GEM depletion or radon accumulation under these stability categories. This shows that changes in advection of GEM from local source/sink regions are not responsible for observed

depletion. Rather, we suggest that the observed depletion results from dry deposition of GEM over terrestrial surfaces. Under increasingly lower capping inversions associated with more stable boundary layers, a near-constant rate of surface deposition would result in greater concentration drops within the boundary layer, consistent with the observations at ATARS. Turbulent break-up of the nocturnal boundary layer at sunrise is also consistent with the rebound of GEM concentrations and drop in diurnal-component radon observed at this time.

Wet season diurnal-component radon values are lower than in the dry season, which fits with wind profile and fetch-component radon data showing greater influence of oceanic fetch. Furthermore, during the wet season rates of radon emission may be reduced in saturated soils, as reduction of pore space inhibits upward mobility to the point where radon within the soil will undergo radioactive decay before reaching the surface (Griffiths et al., 2010). During the wet season, well-mixed and moderately-mixed categories are more indicative of the influence of ocean fetch than stability, as evidenced by wind directions

of $273 \pm 8°$ for these two categories. For weakly-mixed and stable categories, wind direction shifts southerly and easterly throughout the evening, from an oceanic fetch to a terrestrial fetch. It is not until this shift in wind direction occurs that GEM depletion is observed, once again suggesting that these depletion events are due to terrestrial deposition fluxes under nocturnal capping inversion layers. Such a phenomenon would have a significant impact on our understanding of long-range transport of mercury, implying that this transport is due to a "multi-hop" or prompt recycling process of surface deposition and subsequent

reemission, rather than continuous transport over long distances (Selin, 2009). Future research into these depletion events will be undertaken by using measurements of radon fluxes from soils in the region to infer nocturnal GEM fluxes in a manner similar to Obrist et al. (2006).

### 3.2.1   Long-range transport

With seasonal changes in the latitudinal position of the ITCZ, ATARS is periodically located north of the atmospheric equator

(Hamilton et al., 2008) and so the possibility of interhemispheric transport to the site was also of interest. Figure 2 shows the GEOS-Chem output for NH-released tracer concentrations at ATARS. Throughout most of the year — and consistently through the dry season — this value remains low, indicating that the site is far enough below the atmospheric equator to not be affected by transport of NH air. However, there are notable periods when this tracer value increases, along with coincident GEM increases. We arbitrarily defined air masses at the site to be significantly influenced by northern hemisphere air (herein termed





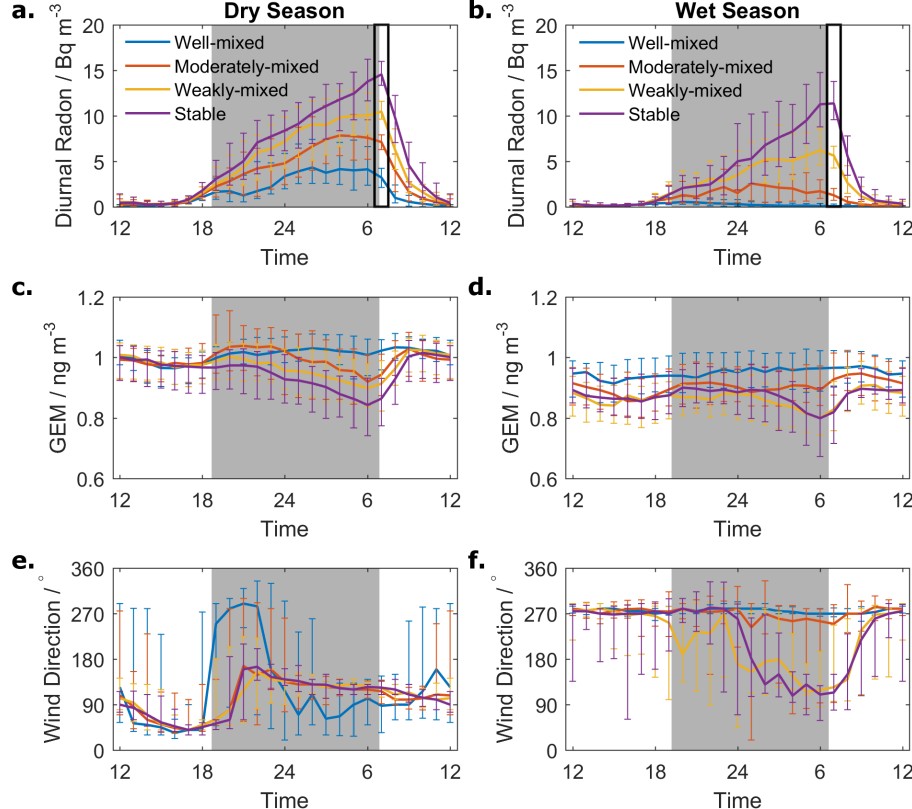

**Figure 4.** Diurnal composites of hourly radon (**a,b**), GEM (**c,d**) and wind direction (**e,f**) for (left) dry season data and (right) all wet season data. Shading denotes median sunset/sunrise times for each season. Data have been split into stability categories based on diurnal-component radon quartiles at sunrise (marked in top panels). Lines are median values and error bars indicate IQRs.

"NH wet season") when the ratio of NH tracers to SH tracers was greater than 0.5 (ratio not shown). Under this definition, ATARS saw 13 NH-influenced days over three distinct periods, all during the wet season and indicated in the lower panel of Fig. 2. Hereafter, wet season data that exclude these periods of NH influence are termed "SH wet season".

The normalised frequency distribution of NH wet season GEM data is compared against those of dry season and SH wet season data in Fig. 5. Mean values for each were 1.08 ng m$^{-3}$, 0.97 ng m$^{-3}$ and 0.90 ng m$^{-3}$, respectively. The differences between these means were small but significant; Student's t-tests showed the minimum differences between the 95 % confidence interval of each mean to be 0.10 ng m$^{-3}$ (NH wet – Dry) and 0.07 ng m$^{-3}$ (Dry – SH wet). Comparison with log-normal probability density functions for other GMOS sites over the years 2013–14 (Fig. 4, Sprovieri et al., 2016) shows that GEM data sampled at ATARS are more closely related to those from other southern hemisphere sites, rather than tropical or northern hemisphere sites. This is likely due to the location of ATARS within the Maritime Continent — a region of high variability in the latitudinal position of the ITCZ — and its southerly latitude that places it outside this range and hence within the atmospheric southern hemisphere for most of the year.



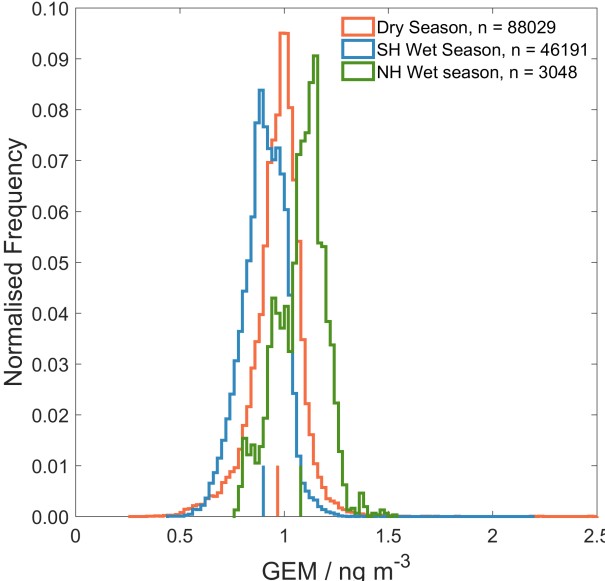

**Figure 5.** Normalised frequency for all 5-minute GEM data, split into dry season, SH wet season and NH wet season. Vertical lines at bottom of figure indicate mean values.

Air mass source transport to ATARS across seasons was further investigated using 5-day HYSPLIT back trajectories. For NH-influenced air masses, use of 5-day trajectories and the geographic equator was found to be a poor predictor of NH influence at this site, with only 1.2 % of these trajectories originating from within the geographical NH. This is likely due to the significant disconnect between the geographical and meteorological equators over the Maritime Continent during the wet

season. As such, 10-day back trajectories were calculated for these periods. Figure 6 shows median, 10th and 90th percentile GEM-weighted trajectory coordinates for 0.5° x 0.5° grid cells. During the dry season (top row), the influence of persistent high pressure cells across the Australian continent can be seen, with most air parcels flowing over central and north-eastern Australia. Changes to air mass source regions are seen with the southward movement of the ITCZ and associated low pressure cells that characterise the SH wet season (centre row). The differing GEM concentration distributions between the two seasons

outlined earlier are further apparent in these two figures. For NH-influenced air masses (bottom row), this analysis shows that most air masses — particularly those with the highest GEM concentrations — passed over the Indonesian archipelago. North of this, air masses moved over the South China Sea or Western Pacific Ocean, with little influence from terrestrial South East Asia. Given that Indonesia's population is greater than 250 million and its biomass burning season coincides with the Australian monsoon, it is likely that the observed increases in GEM concentrations in NH-influenced air masses are more indicative of

anthropogenic or biomass GEM source influence from the Indonesian archipelago than the northern hemisphere background source pool. Further investigation using chemical transport and mercury emission modelling is needed. Regardless, the current analysis shows that ATARS does observe air masses of northern hemisphere origin and that measurements of GEM and other





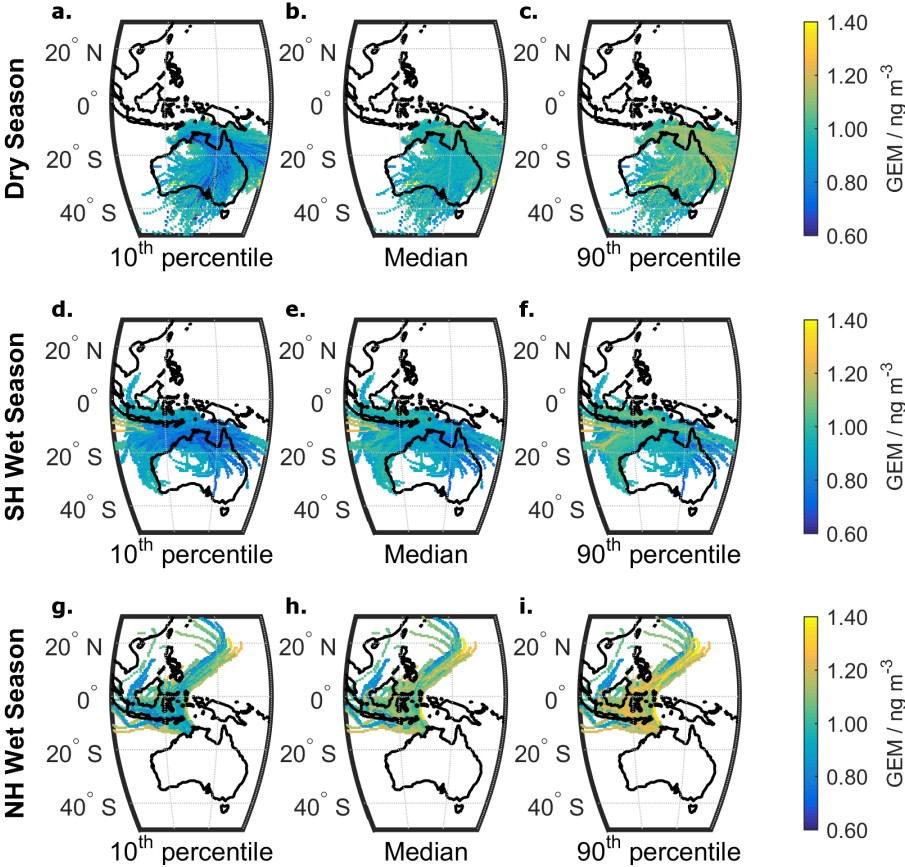

**Figure 6.** 10th percentile (left), median (centre) and 90th percentile (right) of hourly GEM-weighted HYSPLIT trajectories for 0.5° x 0.5° grid squares. **a–c** are for dry season data, **d–f** for SH wet season data and **g–i** for NH wet season data. NH wet season map created using 10-day back trajectories, all others using 5-day trajctories.

atmospheric species during these periods may help to assess the effectiveness of transport models investigating hemispheric air exchange associated with movement of the atmospheric equator.

## 4  Conclusions

We present here the first two years of ongoing measurements of GEM taken in tropical Australia. Comparison with other
5  Australian datasets suggests that a latitudinal gradient of GEM exists across the continent, with higher values towards the equator. Air masses from the northern hemisphere were shown to intermittently impact the tropical site ATARS, with associated increases in GEM. Generally, the concentrations seen at ATARS were indicative of southern hemisphere rather than tropical air, as determined by comparison with other GMOS monitoring stations around the globe.



Seasonal variation in GEM was observed, with higher values observed in the tropical dry season compared to the wet. Spikes in GEM associated with biomass burning in the region were measured, taking place during the mid- to late-dry season. Wet season GEM showed a decreasing trend throughout 2014–15; this was apparent though not as pronounced in the drier 2015–16 season. The cessation of this downward trend coincides with shifts of air mass source regions from oceanic to terrestrial,
however the reverse is not the case for the onset of this trend. It is likely that precipitation rainout or aqueous-phase oxidation of GEM is responsible for this observed downward trend. Continued monitoring and wet deposition data may help to explain these seasonal features.

Daily cycles in GEM were observed at the site, characterised by nocturnal decreases in concentration followed by rapid increases after sunrise, then further decreases throughout the day. Differences in these daily trends between wet and dry
seasons, along with associated changes in wind direction and stability, suggest that these nocturnal depletions are related to dry deposition of GEM over terrestrial surfaces under increasingly stable boundary layers. Such a phenomenon would have a significant impact on our understanding of long-range transport of mercury, implying that this transport is due to a prompt recycling process of surface deposition and subsequent re-emission, rather than continuous transport over long distances.

Currently, multi-annual atmospheric mercury datasets for tropical and SH sites are rare, impacting the skill of regional and
global models designed to further our understanding of the natural mercury cycle and its potential impacts on human and environmental health. The value of measurements such as these is in comparisons with other similar measurements around the globe. As such, the addition of this site to monitoring networks such as the Global Mercury Observation System (GMOS) or the Asia Pacific Mercury Monitoring Network (APMMN) is important in achieving greater understanding of the mercury cycle, as it is currently only one of two monitoring sites located in the tropical eastern hemisphere.

Article 19 of the Minamata Convention commits parties to develop and improve anthropogenic mercury inventories; efforts to monitor mercury and mercury compounds in environmental media; and modelling of mercury transport (including long-range transport and deposition), transformation and fate in a range of ecosystems. ATARS is uniquely positioned to enhance the information required for these monitoring and modelling activities.

## 5   Data availability

GEM data used for this publication are available from the GMOS data repository (http://gmos.eu/sdi/). Weather data are collected and supplied by the Australian Bureau of Meteorology (http://www.bom.gov.au/climate/data-services/).

*Competing interests.* The authors declare that they have no conflict of interest.

*Acknowledgements.* The authors would like to thank Mark Cohen for his assistance with HYSPLIT modelling and for Chris Holmes for supplying code and assistance for GEOS-Chem tracer modelling. This research was undertaken with the assistance of resources provided
at the NCI National Facility systems at the Australian National University through the National Computational Merit Allocation Scheme supported by the Australian Government.





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



**Table 1.** Annual, seasonal and monthly mean, standard deviation and count for 5-minute GEM samples between June 2014 and June 2016. Wet season values calculated from hydrological years beginning in reported year.

| | | Year | Dry | Wet | Jan | Feb | Mar | Apr | May | Jun | Jul | Aug | Sep | Oct | Nov | Dec |
|---|---|---|---|---|---|---|---|---|---|---|---|---|---|---|---|---|
| 2014 | Mean | 1.02 | 1.04 | 0.90 | — | — | — | — | — | 0.99 | 1.02 | 1.08 | 1.08 | 1.03 | 0.95 | |
| | Std. Dev. | 0.10 | 0.10 | 0.12 | — | — | — | — | — | 0.07 | 0.09 | 0.10 | 0.16 | 0.13 | 0.07 | 0.08 |
| | Count | 34734 | 25060 | 24707 | — | — | — | — | — | 2265 | 8421 | 1413 | 2592 | 3845 | 8238 | 7960 |
| 2015 | Mean | 0.93 | 0.94 | 0.93 | 0.92 | 0.79 | 0.76 | 0.82 | 0.89 | 0.95 | 0.96 | 1.00 | 0.99 | 1.01 | 0.99 | 0.96 |
| | Std. Dev. | 0.12 | 0.12 | 0.11 | 0.10 | 0.11 | 0.07 | 0.15 | 0.11 | 0.08 | 0.09 | 0.09 | 0.07 | 0.06 | 0.15 | 0.09 |
| | Count | 72060 | 54071 | 23649 | 8184 | 4830 | 2447 | 5551 | 8848 | 8105 | 6964 | 5613 | 2076 | 5090 | 7636 | 6716 |
| 2016 | Mean | 0.92 | — | — | 0.94 | 0.91 | 0.88 | — | — | — | — | — | — | — | — | — |
| | Std. Dev. | 0.11 | — | — | 0.11 | 0.12 | 0.08 | — | — | — | — | — | — | — | — | — |
| | Count | 21576 | — | — | 8438 | 7631 | 5507 | — | — | — | — | — | — | — | — | — |