# Peer review of "Atmospheric mercury in the southern hemisphere tropics: seasonal and diurnal variations and influence of inter-hemispheric transport"

_Atmospheric Chemistry and Physics, 2017_

## Referee Comment (RC1) · Anonymous Referee #1 · 7 May 2017

The authors present data of two year mercury and some other ancillary measurements at a site in northern Australia near the ITCZ. They compare their data with other measurements in southern hemisphere and analyse their seasonal and diurnal variations.

As one of the very few long-term measurements in the tropics these measurements are very valuable, as is their analysis. The paper is generally well ordered and written. The subject fits the scope of ACP and the paper should be published. I recommend the publication after consideration of the comments listed below:

Introduction and text: The authors declare night time deposition events as a strong evidence for a "multi-hop" mercury transport model. However, they present no evidence for day time reemissions necessary for being able to talk about "hops". Without such reemissions the "multi-hop" transport model remains a lyrics rather than a model. There are other problems with this model: Isn't there a discrepancy between a "multi-hop" model and an atmospheric GEM lifetime of 6 – 12 months mentioned in the "Introduction"? Or between a "multi-hop" model and chapter 3.2.1 on long-range transport?

Section 2.2: Please state the standard conditions (pressure, temperature) at which the Hg concentrations are reported.

Section 3.1: Are the latitudinal differences statistically significant?

Page 2, line 17: "existence"?

Page 4, line 7: Reference "Köppen Aw" is missing in the list of references.

Page8, line 9: Andreae and Merlet (2001) is a review article citing largely work by others. They did not determine the mercury emission factor from biomass burning. Please reword or cite the original work.

Page 8, line 23: a net sink

Page 9, line 23: Year of the reference "Howard et al."?

Page 11, caption of Fig.4: Median values are usually points – please reword. What is IQR?

Page 11, line 5: Mean values should be given with their standard deviations or errors and the number of measurements.

Figure 5: Vertical line at bottom for NH wet season merges with the blue line. Please make it more distinct.

---

## Referee Comment (RC2) · Anonymous Referee #2 · 19 May 2017

Howard et al. present two years of GEM measurements at a tropical station. Very few mercury measurements have been made in this region and this study partly fills the data gaps. This paper is well written and is within the scope of this journal. I have several minor comments: (1) Introduction, first paragraph: it should be made clear that the mercury emissions here refer to those emitted into the atmosphere; (2) Introduction: Soerensen et al. 2014 ES&T Elemental Mercury Concentrations and Fluxes in the Tropical Atmosphere and Ocean analyzed GEM fluxes in the tropical ocean and is a relevant reference for the third paragraph; (3) Page 4, line 10: "relative" humidity; (4) Page 6, Section 3.1: Slemr et al. 2015 ACP Comparison of mercury concentrations measured at several sites in the Southern Hemisphere suggested that

the systematic uncertainty among the GEM measurements at different stations is about 0.1 ng/m3. I am not sure whether the 9% difference between the ATARS station and Cape Grim really indicates a latitudinal gradient. (5) Page 8, lines 22-24: The Ocean is commonly a sink of total atmospheric mercury, but is commonly a net source of GEM (the measured mercury specie). (6) Page 9, line 23, "Howard et al." typo; (7) As this station also has data of other chemical species such as aerosol and ozone, including these data would highly facilitate the data analysis (for example on the effect of biomass burning on GEM).

---

## Author Comment (AC1) · 29 Jul 2017

We thank the anonymous reviewer for the positive comments and for the very helpful suggestions. These have made many points in the paper much clearer. Please find below the comments and responses, numbered by individual comment. Within each entry we provide a) the reviewer comment, b) our response, and c) changes made to the manuscript. (Note: the attached supplement shows this in a much clearer format).

1. a. The authors declare night time deposition events as a strong evidence for a "multi-hop" mercury transport model. However, they present no evidence for day time reemissions necessary for being able to talk about "hops". b. This is an important

point and we thank the reviewer for highlighting this. We agree that re- emission of GEM following the nocturnal depletion was not covered in the text and have revised it accordingly. Emission of GEM in the morning precedes the break-up of the nocturnal boundary layer (as evidenced by diurnal-component radon), suggesting emission from the surface. As further evidence of emission, in the early morning GEM concentrations overshoot what could be considered the background GEM concentration, observed in the mid-afternoon when boundary layer mixing is at its greatest. It is reasonable to hypothesise that this early morning emission of GEM is partly made up of the relatively volatile fraction of newly-deposited mercury, based on experience in Arctic AMDEs. Further, observed net GEM fluxes during similar depletion events at another Australian site showed that these depletions were not significant long-term sinks of GEM. We have reworded this section to make this clearer. c. Page 10, Line 9: Reworded most of Section 3.2, through to Page 12, Line 16.

2. a. Isn't there a discrepancy between a "multi-hop" model and an atmospheric GEM lifetime of 6 – 12 months mentioned in the "Introduction"? b. We don't believe there is a discrepancy, as the prompt recycling process would not be resolved in the methods used to estimate GEM lifetime. We have reworded this section to define atmospheric lifetime as the mean time before which GEM is permanently removed from the atmosphere. As, under a "multi-hop" model, deposited GEM is rapidly re-emitted, this does not represent a permanent sink and would not affect calculation of mean atmospheric lifetime as determined from mass-balance approaches. c. Page 2, Line 23: Removed "The low atmospheric . . . inter-hemispheric transport processes." Page 2, Line 20: Inserted "The low atmospheric reactivity and low solubility of the elemental form (GEM) results in low wet/dry deposition rate and scavenging of GEM from the atmosphere. These attributes result in atmospheric transport being the dominant distribution mechanism through the environment, with long-range transport possible across hemispheric scales. Differences in background atmospheric mercury concentrations between the hemispheres are hence dependent on emission rates, deposition rates, inter- hemispheric transport processes, and atmospheric mercury lifetimes. The atmospheric lifetime is defined here as the mean time after emission that GEM is removed from the atmosphere (Lindberg et al., 2007) and is estimated from mass–balance approaches utilising hemispheric background concentration and source/sink data (e.g. Slemr et al., 1985). The atmospheric lifetime of GEM is currently estimated at 5–12 months (Holmes et al., 2006; Selin et al., 2007; Holmes et al., 2010; Horowitz et al., 2017)."

3. a. Or between a "multi-hop" model and chapter 3.2.1 on long-range transport? b. We agree with the reviewer here that the wording used seemed to attempt to replace the model of long-range transport with one of continual deposition and reemission. This was not intended, as differing processes can take place within the planetary boundary layer and free troposphere. As re-emission is difficult to differentiate from emission in GEM flux studies it is possible that evidence for this prompt recycling process has not been clear in the past. Our intention is to highlight, through the evidence of the nocturnal depletion events, that surface deposition/re-emission may be more important to atmospheric mercury cycling than previously believed. We have reworded multiple sections to make this clearer. c. Page 10, Line 23: Removed "Such a phenomenon . . . over long distances." Page 12, Line 12: Inserted "It is important to note that, due to inhibited mixing at the top of the nocturnal boundary layer, the extent of this depletion is limited to within tens to hundreds of metres above the surface. Beyond this, movement of free-tropospheric air continues to enable long-range transport of GEM. Nevertheless, were this phenomenon of rapid, bidirectional exchange with the surface to occur it would have a significant impact on our understanding of atmospheric mercury transport as it can impact the relative importance of intermediate and regional-scale sources, as well as expected time scales for observed decreases in environmental mercury following actions proposed under the Minamata Convention (Lindberg et al., 2007)." Page 1, Line 16: Removed "These cycles provide . . . over long distances." Page 1, Line 16: Inserted "These cycles provide strong further evidence supportive of a "multi-hop" model of GEM cycling, characterised by multiple surface depositions and re- emissions, in addition to long-range transport through the atmosphere." Page 14, Line 11: Removed "Such a phenomenon . . . over long distances." Page 15, Line 14: Inserted "Analyses using diurnal-component radon suggest the rapid increases around sunrise are partly due to volatilisation of newly-deposited mercury, such as seen in other NAMDEs and Arctic AMDEs. The extent of this multi-hop phenomenon may be widespread, which would have a significant impact on our understanding of atmospheric mercury transport, the delivery of atmospheric mercury to the environment, and the legacy of anthropogenic emissions of mercury."

4. a. Section 2.2: Please state the standard conditions (pressure, temperature) at which the Hg concentrations are reported. b. This should have been included and was overlooked. c. Page 5, Line 5: Inserted "Reference volumes are reported at 1 atm and 0 âŮęC."

5. a. Section 3.1: Are the latitudinal differences statistically significant? b. The values themselves are, however additional uncertainty as a result of the instrumentation and differences in sampling periods suggests that this is not the case. c. Page 7, Line 10: Added "These differences are statistically significant (Student's t-test, $p < 0.0001$), though differences in the sampling periods introduces additional uncertainty due to seasonal variation at the sites. Further, an analysis of systematic instrument uncertainty for the Tekran 2537 by Slemr et al. (2015) showed this to be ∼10 %."

6. a. Page 2, line 17: "existence"? b. The wording here has been altered. c. Page 2, Line 17: Removed "resulting from its existence . . . environmental conditions."

7. a. Page 4, line 7: Reference "Köppen Aw" is missing in the list of references. b. "Köppen Aw" was intended as the category type reported by Peel et al., though this was not made clear. c. Page 4, line 7: Changed reference to "Köppen category Aw, as reported by Peel et al., 2007".

8. a. Andreae and Merlet (2001) is a review article citing largely work by others. They did not determine the mercury emission factor from biomass burning. Please reword or cite the original work. b. The use of "determined" was incorrect and unintended. c. Page 8, Line 5: Changed "determined by Andreae and Merlet (2001)" to "reported by

Andreae and Merlet (2001, and references within)"

9. a. Page 8, line 23: a net sink b. In response to this comment and a similar one from Reviewer 2, we have altered the text to better describe air–sea GEM exchange. c. Page 8, Line 23: Removed "Air-sea exchange of GEM . . . from soil and vegetation." Page 9, Line 11: Inserted "Air-sea exchange of mercury is complex, with the ocean generally considered a net sink for atmospheric mercury (Mason and Sheu, 2002; Song et al., 2015). Reduction of mercury within the photolytic zone can give rise to increased concentrations of elemental mercury and hence evasion of GEM to the atmosphere (Soerensen et al., 2014). Terrestrial surfaces are also commonly sources of GEM; Nelson et al. (2012) modelled terrestrial mercury emission fluxes over Australia that were generally between 8 and 44 $\mu$g m-2 a-1 from soil and vegetation. Figure 3 does not show a strong difference in concentration distributions between the two source regions."

10. a. Page 9, line 23: Year of the reference "Howard et al."? b. This was an error; the correct reference should have been Howard and Edwards (2017). c. See 1c.

11. a. Page 11, caption of Fig.4: Median values are usually points – please reword. What is IQR? b. The shading shows nocturnal periods, hence the wording should have noted that the shading edges are sunset/sunrise times. IQR refers to inter-quartile range and was not defined. c. Changed caption to "Edges of shading denote median sunset/sunrise times for each season. Data have been split into stability categories based on diurnal-component radon quartiles at sunrise (marked in top panels). Lines are median values and error bars indicate inter-quartile ranges."

11. a. Page 11, line 5: Mean values should be given with their standard deviations or errors and the number of measurements. b. These have been included. c. Page 12, Line 30: Included "Mean values for each were 1.08 $\pm$ 0.12 ng m-3 (n = 3048), 0.97 $\pm$ 0.13 ng m-3 (n = 81073) and 0.90 $\pm$ 0.10 ng m-3 (n = 46191), respectively."

12. a. Figure 5: Vertical line at bottom for NH wet season merges with the blue line.

Please make it more distinct. b. This has been fixed. c. Altered Figure 5.

Please also note the supplement to this comment:
https://www.atmos-chem-phys-discuss.net/acp-2017-307/acp-2017-307-AC1-
supplement.pdf
* * *
[Figure]

[Figure]

**Fig. 1.**

---

## Author Comment (AC2) · 29 Jul 2017

We are grateful to the anonymous reviewer for their thoughtful comments and for their positive comments regarding the manuscript. These comments have helped to sharpen a number of points within. Please find below the comments and responses, numbered by individual comment. Within each entry we provide a) the reviewer comment, b) our response, and c) changes made to the manuscript. (Note: the formatting is much clearer in the attached supplement).

1. a. Introduction, first paragraph: it should be made clear that the mercury emissions here refer to those emitted into the atmosphere b. This has been fixed accordingly. c.

[Figure]

Page 2, Line 5: Inserted: "to the atmosphere"

2. a. Introduction: Soerensen et al. 2014 ES&T Elemental Mercury Concentrations and Fluxes in the Tropical Atmosphere and Ocean analyzed GEM fluxes in the tropical ocean and is a relevant reference for the third paragraph b. Included this reference in the introduction and in Section 3.1 (see 5c). c. Page 3, Line 8: Inserted "Soerensen at al., 2014."

3. a. Page 4, line 10: "relative" humidity b. This has been changed. c. Page 4, Line 11: Inserted "relative".

4. a. Page 6, Section 3.1: Slemr et al. 2015 ACP Comparison of mercury concentrations measured at several sites in the Southern Hemisphere suggested that the systematic uncertainty among the GEM measurements at different stations is about 0.1 ng/m3. I am not sure whether the 9% difference between the ATARS station and Cape Grim really indicates a l latitudinal gradient. b. This is an important point raised by the reviewer. Although the values are statistically different, there are other uncertainties that bring into question this significance, including the systematic instrumental uncertainty raised by the reviewer. We have altered the text to highlight these uncertainties. c. Page 7, Line 10: Added "These differences are statistically significant (Student's t-test, p < 0.0001), though differences in the sampling periods introduces additional uncertainty due to seasonal variation at the sites. Further, an analysis of systematic instrument uncertainty for the Tekran 2537 by Slemr et al. (2015) showed this to be ∼10 %."

5. a. Page 8, lines 22-24: The Ocean is commonly a sink of total atmospheric mercury, but is commonly a net source of GEM (the measured mercury species). b. We thank the reviewer for pointing out this distinction and have adjusted the text accordingly. c. Page 8, Line 23: Removed "Air-sea exchange of GEM . . . from soil and vegetation." Page 9, Line 11: Inserted "Air-sea exchange of mercury is complex, with the ocean generally considered a net sink for atmospheric mercury (Mason and Sheu, 2002; Song

et al., 2015). Reduction of mercury within the photolytic zone can give rise to increased concentrations of elemental mercury and hence evasion of GEM to the atmosphere (Soerensen et al., 2014). Terrestrial surfaces are also commonly sources of GEM; Nelson et al. (2012) modelled terrestrial mercury emission fluxes over Australia that were generally between 8 and 44 $\mu$g m-2 a-1 from soil and vegetation. Figure 3 does not show a strong difference in concentration distributions between the two source regions."

6. a. Page 9, line 23, "Howard et al." typo b. The typo has been edited. c. Page 10, Line 4: Changed "Howard et al." to "Howard and Edwards (2017)".

7. a. As this station also has data of other chemical species such as aerosol and ozone, including these data would highly facilitate the data analysis (for example on the effect of biomass burning on GEM). b. The inclusion of such data would quite likely facilitate additional analyses, however they are not currently evaluated to the standard required for publication. When these data become available, future analyses will be undertaken. c. No changes made.

Please also note the supplement to this comment:
https://www.atmos-chem-phys-discuss.net/acp-2017-307/acp-2017-307-AC2-supplement.pdf

---

## Author Comment (AC3) · 29 Jul 2017

**Atmospheric mercury in the southern hemisphere tropics: seasonal and diurnal variations and influence of inter-hemispheric transport**

Dean Howard[1], Peter F. Nelson[1], Grant C. Edwards[1], Anthony L. Morrison[1], Jenny A. Fisher[2,3], Jason Ward[4], James Harnwell[4], Marcel van der Schoot[4], Brad Atkinson[5], Scott D. Chambers[6], Alan D. Griffiths[6], Sylvester Werczynski[6], and Alastair G. Williams[6]

[1]Department of Environmental Sciences, Macquarie University, Sydney, New South Wales, 2109, Australia
[2]Centre for Atmospheric Chemistry, School of Chemistry, University of Wollongong, Wollongong, New South Wales, 2552, Australia
[3]School of Earth & Environmental Sciences, University of Wollongong, Wollongong, New South Wales, 2552, Australia
[4]Oceans and Atmosphere Flagship, Commonwealth Science and Industrial Research Organisation, Aspendale, Victoria, 3195, Australia
[5]Darwin Research Station, Bureau of Meteorology, Darwin, Northern Territory, 0810, Australia
[6]Institute for Environmental Research, Australian Nuclear Science and Technology Organisation, Sydney, New South Wales, 2232, Australia

*Correspondence to:* Dean Howard (dean.howard@mq.edu.au)

**Abstract.** Mercury is a toxic element of serious concern for human and environmental health. Understanding its natural cycling in the environment is an important goal towards assessing its impacts and the effectiveness of mitigation strategies. Due to the unique chemical and physical properties of mercury, the atmosphere is the dominant transport pathway for this heavy metal, with the consequence that regions far removed from sources can be impacted. However, there exists a dearth of long-term monitoring of atmospheric mercury, particularly in the tropics and southern hemisphere. This paper presents the first two years of gaseous elemental mercury (GEM) measurements taken at the Australian Tropical Atmospheric Research Station (ATARS) in northern Australia, as part of the Global Mercury Observation System (GMOS). Annual mean GEM concentrations determined at ATARS ($0.95 \pm 0.12$ ng m$^{-3}$) are consistent with recent observations at other sites in the southern hemisphere. Comparison with GEM data from other Australian monitoring sites suggests a concentration gradient that decreases with increasing latitude. Seasonal analysis shows that GEM concentrations at ATARS are significantly lower in the distinct wet monsoon season than in the dry season. This result provides insight into alterations of natural mercury cycling processes as a result of changes in atmospheric humidity, oceanic/terrestrial fetch and convective mixing, and invites future investigation using wet mercury deposition measurements. Due to its location relative to the atmospheric equator, ATARS intermittently samples air originating from the northern hemisphere, allowing an opportunity to gain greater understanding of inter-hemispheric transport of mercury and other atmospheric species. Diurnal cycles of GEM at ATARS show distinct nocturnal depletion events that are attributed to dry deposition under stable boundary layer conditions. These cycles provide strong further evidence supportive of a "multi-hop" model of GEM cycling, characterised by multiple surface depositions and re-emissions, in addition to long-range transport through the atmosphere.

**1 Introduction**

Mercury (Hg) is a toxic element that has natural and anthropogenic sources, sinks and cycles within the environment. Human activities such as gold mining and biomass/fossil fuel combustion have perturbed the natural cycling of mercury through the addition of mercury emissions, which are re-deposited from the atmosphere to land, vegetation and water bodies. It is estimated that currently anthropogenic emissions to the atmosphere increase the global atmospheric mercury pool by 1960 tonnes annually, a value that represents 30 % of estimated mercury emissions, with the remainder emitted from natural geological sources (10 %) or re-emitted from stores of previously-deposited mercury (60 %). These mercury emission estimates are subject to large uncertainties (AMAP/UNEP, 2013; UNEP, 2013). That anthropogenic mercury sources now exceed those from natural sources on a global scale is of concern for both human and environmental health. Evidence suggests these additional sources are leading to increased concentrations of mercury in the oceans and in marine animals, with the consequence that bioaccumulation of toxic methylmercury within aquatic food chains has also increased (Mason et al., 2012; UNEP, 2013). There exists a significant pathway for methylmercury transfer to humans, as it is estimated that more than 100 million tonnes of fish are eaten world-wide each year and fish provide two and a half billion people with at least 20 % of their protein intake. Mercury in this latter form can seriously threaten human health through impacts on the development of foetuses and young children. In response to this threat, the United Nations Environment Programme (UNEP) has developed the Minamata Convention on Mercury, which is expected to be ratified in 2017.

The global cycling of mercury is unique amongst metals, as within Earth's atmosphere 90 to 99 % of mercury is found as gaseous elemental mercury (GEM), with the remaining portion composed of operationally-defined gaseous oxidised mercury (GOM) and particulate-bound mercury (PBM) — collectively known as reactive mercury (RM) (Gustin et al., 2013). The low atmospheric reactivity and low solubility of the elemental form (GEM) results in low wet/dry deposition rates and scavenging of GEM from the atmosphere. These attributes result in atmospheric transport being the dominant distribution mechanism through the environment, with long-range transport possible across hemispheric scales. Differences in background atmospheric mercury concentrations between the hemispheres are hence dependent on emission rates, deposition rates, inter-hemispheric transport processes, and atmospheric mercury lifetimes. The atmospheric lifetime is defined here as the mean time after emission that GEM is removed from the atmosphere (Lindberg et al., 2007) and is estimated from mass-balance approaches utilising hemispheric background concentration and source/sink data (e.g. Slemr et al., 1985). The atmospheric lifetime of GEM is currently estimated at 5–12 months (Holmes et al., 2006; Selin et al., 2007; Holmes et al., 2010; Horowitz et al., 2017).

[revised manuscript text omitted]

The vegetation classification is savannah with coarse grasses and scattered tree growth immediately surrounding the site. Burning of the grassed areas occurs frequently, with a fire return interval of 1–2 years. Direct mercury analysis (see Edwards and Howard, 2013, for methodology) of vegetation within 500 m of the station gave total mercury concentrations of $7.23 \pm 0.37$ $\mu$g kg$^{-1}$ ($n = 18$) for grass and $21.09 \pm 3.79$ $\mu$g kg$^{-1}$ ($n = 9$) for tree litter. Sampling of soils in the same locations gave total mercury concentrations of $9.14 \pm 0.58$ $\mu$g kg$^{-1}$ ($n = 18$) in grassed areas and $26.49 \pm 3.31$ $\mu$g kg$^{-1}$ ($n = 9$) under forest canopy, confirming that soils in the area are categorised as background ($< 100$ $\mu$g kg$^{-1}$; Gustin et al., 2006). Sampling was undertaken in the early dry season, approximately 10–12 months after the last grass fire.

[revised manuscript text omitted]
. These differences are statistically significant (Student's t-test, $p < 0.0001$), though differences in the sampling periods introduces additional uncertainty due to seasonal variation at the sites. Further, an analysis of systematic instrument uncertainty for the Tekran 2537 by Slemr et al. (2015) showed this to be ∼10 %. A latitudi-nal gradient within the southern hemisphere was more generally seen in median annual GEM concentrations for GMOS sites in 2013–14, based on data from 5 sites (Sprovieri et al., 2016). GEM measurements at ATARS were coincident with those
15   reported by Sprovieri et al. (2016) for only the latter 6 months of 2014, a period spanning the late dry season and early wet season. Concentrations during this period were $1.02 \pm 0.10$ ng m$^{-3}$ — higher than the overall mean at ATARS, though still lower than mean values reported for other tropical GMOS sites.

[revised manuscript text omitted]

35   the hour before diurnal-component radon signals the break-up of the nocturnal boundary layer. In the absence of changes to

[Figure]

**Figure 4.** Diurnal composites of hourly radon (**a,b**), GEM (**c,d**) and wind direction (**e,f**) for (left) dry season data and (right) all wet season data. Edges of shading denote median sunset/sunrise times for each season. Data have been split into stability categories based on diurnal-component radon quartiles at sunrise (marked in top panels). Lines are median values and error bars indicate inter-quartile ranges.

advection or entrainment, this suggests emission of GEM from the surface. Furthermore, for stability categories where GEM depletion has taken place, daytime GEM concentration peaks at around 10:00 before decreasing to a minimum at around 15:00, where low radon values indicate the strongest turbulent mixing with free-tropospheric air. This "overshoot" of GEM in the early morning also cannot be explained by entrainment and, at least in the dry season, by changes to fetch. Early-morning
5   GEM emission would likely be from the most readily volatile surface mercury, released under low-light conditions (shading denotes the period between geometric sunset/sunrise and so astronomical twilight will begin up to 75 minutes prior to the shaded edge). We propose that this initial release of GEM is volatilised from the reduction of mercury deposited overnight, as it has been shown that the most recently-deposited mercury during AMDEs is preferentially released due to photochemical reactions (Sherman et al., 2010).
10      Previous studies have shown that surface GEM fluxes over soils with mercury concentration at background levels are generally bi-directional, with little controlling influence from soil mercury concentration (Agnan et al., 2016, and references within). Correlations with solar radiation and air temperature tend to lead to emission fluxes throughout the day and deposition or nearzero flux overnight. Howard and Edwards (2017), whilst undertaking micrometeorological measurements of surface GEM fluxes over a background mercury substrate grassland, observed nocturnal atmospheric mercury depletion events (NAMDEs) similar to the ones seen at ATARS. They attributed these events to enhanced nocturnal deposition of GEM under shallow, stable boundary layers. Enhancements in morning GEM emission were seen in days following the depletion events, similarly provid-
5  ing evidence for volatilisation of recently-deposited mercury. Further, cumulative GEM exchange over the 20-day study was near zero, highlighting the short-lived nature of this nocturnal GEM sink. This result, and the radon-based analyses presented earlier, provide strong evidence for a "multi-hop" process of atmospheric transport.

NAMDEs have also been observed in the northern hemisphere, in a range of ecosystems ranging from coastal to forested (Mao et al., 2008; Engle et al., 2010; Fu et al., 2016). Mao et al. (2008) attributed 70 % of their observed depletion to surface
10  deposition and Fu et al. (2016) provided modelling evidence showing that stable boundary layers of height 100 m can be completely depleted of GEM due to deposition processes. The pervasiveness of NAMDEs across multiple ecosystems, and their pervasiveness throughout the ATARS time series across all seasons, suggests this multi-hop process is widespread. It is important to note that, due to inhibited mixing at the top of the nocturnal boundary layer, the extent of any nocturnal depletion is limited to within tens to hundreds of metres above the surface. Beyond this, movement of free-tropospheric air continues to
15  enable long-range transport of GEM. Nevertheless, extensive and rapid bi-directional exchange with the surface would have a significant impact on our understanding of atmospheric mercury transport, impacting the relative importance of intermediate and regional-scale sources, as well as expected time scales for observed decreases in environmental mercury following actions proposed under the Minamata Convention (Lindberg 
[revised manuscript text omitted]

Griffiths, A. D., Chambers, S. D., Williams, A. G., and Werczynski, S.: Increasing the accuracy and temporal resolution of two-filter radon–222 measurements by correcting for the instrument response, Atmospheric Measurement Techniques, 9, 2689–2707, doi:10.5194/amt-9-2689-2016, 2016.

Gustin, M. S., Engle, M., Ericksen, J., Lyman, S., Stamenkovic, J., and Xin, M.: Mercury exchange between the atmosphere and low mercury containing substrates, Applied Geochemistry, 21, 1913–1923, doi:10.1016/j.apgeochem.2006.08.007, 2006.

Gustin, M. S., Huang, J., Miller, M. B., Peterson, C., Jaffe, D. A., Ambrose, J., Finley, B. D., Lyman, S. N., Call, K., Talbot, R., Feddersen, D., Mao, H., and Lindberg, S. E.: Do we understand what the mercury speciation instruments are actually measuring? Results of RAMIX, Environmental Science and Technology, 47, 7295–7306, doi:10.1021/es3039104, 2013.

Hamilton, J. F., Allen, G., Watson, N. M., Lee, J. D., Saxton, J. E., Lewis, A. C., Vaughan, G., Bower, K. N., Flynn, M. J., Crosier, J., Carver, G. D., Harris, N. R., Parker, R. J., Remedios, J. J., and Richards, N. A.: Observations of an atmospheric chemical equator and its implications for the tropical warm pool region, Journal of Geophysical Research: Atmospheres, 113, doi:10.1029/2008JD009940, 2008.

Hansen, A. and Gay, D.: Observations of mercury wet deposition in Mexico, Environmental Science and Pollution Research, 20, 8316–8325, doi:10.1007/s11356-013-2012-3, 2013.

Holmes, C., Jacob, D., Corbitt, E., Mao, J., Yang, X., Talbot, R., and Slemr, F.: Global atmospheric model for mercury including oxidation by bromine atoms, Atmospheric Chemistry and Physics, 10, 12 037–12 057, doi:10.5194/acp-10-12037-2010, 2010.

Holmes, C. D. and Prather, M. J.: An atmospheric definition of the equator and its implications for atmospheric chemistry and climate, Nature Geoscience, in press.

Holmes, C. D., Jacob, D. J., and Yang, X.: Global lifetime of elemental mercury against oxidation by atomic bromine in the free troposphere, Geophysical Research Letters, 33, doi:10.1029/2006GL027176, 2006.

Horowitz, H. M., Jacob, D. J., Zhang, Y., Dibble, T. S., Slemr, F., Amos, H. M., Schmidt, J. A., Corbitt, E. S., Marais, E. A., and Sunderland, E. M.: A new mechanism for atmospheric mercury redox chemistry: implications for the global mercury budget, Atmospheric Chemistry and Physics, 17, 6353–6371, doi:10.5194/acp-17-6353-2017, 2017.

Howard, D. and Edwards, G. C.: Mercury fluxes over an Australian alpine grassland and observation of nocturnal atmospheric mercury depletion events, Atmospheric Chemistry and Physics Discussions, 2017.

Lindberg, S., Bullock, R., Ebinghaus, R., Engstrom, D., Fenh, X., Fitzgerald, W., Pirrone, N., Prestbo, E., and Seigneur, C.: A Synthesis of Progress and Uncertainties in Attributing the Sources of Mercury in Deposition, AMBIO: A Journal of the Human Environment, 36, 19–33, doi:10.1579/0044-7447(2007)36[19:ASOPAU]2.0.CO;2, 2007.

[revised manuscript text omitted]

---

## Author Comment (AC4) · 29 Jul 2017

Please find attached revised manuscript following reviewer comments.

Please also note the supplement to this comment:
https://www.atmos-chem-phys-discuss.net/acp-2017-307/acp-2017-307-AC4-supplement.pdf

<hr>